# ‘Boat Quarantine’: Lessons Learned from SARS-CoV-2 Prevention and Control Measures in Fishing Communities in Thailand

**DOI:** 10.3390/ijerph20064816

**Published:** 2023-03-09

**Authors:** Niphattra Haritavorn

**Affiliations:** Faculty of Public Health, Thammasat University (Rangsit Campus), Pathumthani 12120, Thailand; niphattraph@gmail.com; Tel.: +66-2564-4440 (ext. 7426); Fax: +66-2516-2708

**Keywords:** SARS-CoV-2, COVID-19, quarantine, qualitative, communities, fishermen, Thailand

## Abstract

SARS-CoV-2 posed, and continues to pose, a severe threat to life, and for fishermen in Thailand, specific multifaceted quarantine design measures have been required. In response to the SARS-CoV-2 outbreak in Trat province, a community quarantine centre was designed using boats as quarantine facilities. This study examines the implementation of boat quarantine in response to the SARS-CoV-2 pandemic within the fishermen communities in Trat province, Thailand. In-depth interviews with 45 key individuals who have been involved in the control and prevention of SARS-CoV-2 among fishermen in the fishing communities were subjected to a thematic analysis. Boat quarantine was used to separate and restrict contact between fishermen who were exposed to SARS-CoV-2, to determine whether they became sick and to prevent mass infection within the community. Using a boat as a place to self-isolate has become an effective form of quarantine for fishermen. This model has implications for the future of infectious disease control onshore, both while the pandemic continues and after the pandemic comes to an end.

## 1. Introduction

The SARS-CoV-2 pandemic has become the defining global health crisis of the twenty-first century. It was first reported in December 2019 in Wuhan, China. The virus has since spread to almost all countries around the world. An outbreak of SARS-CoV-2 occurred in Thailand around March 2020 and has since spread to all provinces. The number of cases rose in 2021 when vaccination was limited, and more cases were reported. On 14 January 2022, nearly 2,308,615 confirmed cases of SARS-CoV-2 had been reported in Thailand and more than 21,898 people had died, as reported by the Thai government.

Currently, the world is dealing with the SARS-CoV-2 pandemic. In order to control SARS-CoV-2, many countries subsequently implemented emergency measures, including multilateral health surveillance systems [1,2]. Notably, quarantine schemes have been adopted in many countries, such as China, Germany, and the Philippines [3]. A community quarantine scheme against SARS-CoV-2 was implemented in Anhui, that was led by community committees largely made up of public servants. These committees were aimed at identifying residents with a definite exposure history [4]. Moreover, SARS-CoV-2 quarantine schemes for seafarers have been adopted, as Cordreanu et al. [5] noted, when a SARS-CoV-2 outbreak onboard a cruise ship in Australia occurred, it was effectively controlled, as the ship served as a quarantine facility that enforced the process of isolating patients, quarantining exposed individuals, and segregating the onboard crew.

Moreover, community quarantine schemes are designed to meet the needs of the of the communities they serve in order to reduce the likelihood of transmission. Notably, ‘community quarantines’ have been created to cope with many infectious diseases. For example, due to the Ebola outbreak in Africa, the Liberian government set up a community quarantine in Mawah village, Bong county [6]. The Philippine government put into service ‘enhanced community quarantines’ to fight SARS-CoV-2, which have been described as “the implementation of temporary measures imposing stringent limitations on the movement and transportation of people, strict regulations for operating industries, the provision of food and essential services, and the heightened presence of uniformed personnel to enforce community quarantine protocols” [7].

However, the Thai government implemented a range of disease-containment strategies: the closure of public gatherings, testing and treating patients, lockdown of the epicenter, carrying out contact tracing, and limiting travel and social gatherings [8,9]. As SARS-CoV-2 spread and border lockdowns took place, living and working conditions changed following the Thai government’s regulations. Notably, quarantine became the tool that the Ministry of Public Health employed to protect the country, as the government believed that quarantine reduced the risk of spreading the virus in addition to not allowing people to enter the country. Hence, travel restrictions in Thailand in 2020–2021 required that all persons suspected of having SARS-CoV-2 be quarantined for 14 days until no longer infected, especially anyone who had been in contact with infected persons or who came from high-risk areas where SARS-CoV-2 was present, and sick persons were sent to the hospital. In Thailand, a community quarantine model for controlling SARS-CoV-2 entitled “Clean Community Anti COVID-19 (SARS-CoV-2)” was implemented within a crowded community in Bangkok [10]. Although the community quarantine model applied to all populations in the country, the specific model was designed to meet the needs of seafarers.

The maritime quarantine scheme was implemented for seafarers in specific settings. The pre-existing structure (social structure) caused seafarers in distant waters to be vulnerable to the SARS-CoV-2 pandemic [11,12]. The seafarers faced several problems, such as delays at the port, poor working environment, tiny spaces on vessels, frequent travel from vessel to vessel [13], crew changes, restrictions on access to onshore services, and unclear status with respect to vaccine access [14]. The maritime surveillance and quarantining faced the following challenges: the management of active cases onboard; the need to establish physical distancing and other measures to reduce the spread of the disease on the ship; access to pre-employment medical examinations; interaction with onshore staff in ports; crew changes; access to medical, dental, and welfare services in port; reduced possibilities for shore leave; contract extensions; and an increase in mental health issues among seafarers onboard [15]. During the SARS-CoV-2 pandemic, some cruise ships were used as quarantine vessels [16].

Trat, one of the eastern provinces in Thailand, is situated on the border between Cambodia and Thailand. The province shares not only a land border but also a sea border with Cambodia, which makes migration from Cambodia to Thailand easy. Trat is a center of gemstone mining, tropical fruits, and fishery products. A large number of fish are delivered to factories for further processing, and the rest are sold in local markets. Importantly, the fishing industry in Trat has been seriously affected by the SARS-CoV-2 pandemic [17]. In May 2022, there were approximately 14,532 SARS-CoV-2 cases in Trat. So far, almost 56 million people in the province have been vaccinated against SARS-CoV-2. The number of seafarers infected with the disease in Laew Ngob and Khong Yai, where the major fishery port is located, was approximately 700, as of 31 March 2022. Due to the shortage of health facilities between September and December 2021, the peak time of the pandemic, boat quarantine was implemented. The quarantine followed the Ministry of Public Health’s SARS-CoV-2 (COVID-19) guidelines that stated that the official quarantine period was to last 14 days.

Notably, fishermen have been identified as a group with an increased risk of SARS-CoV-2 infection, yet they have been neglected in the research in Thailand. Boat quarantine serves as a good model to study the potential for SARS-CoV-2 to spread within a population in the fishing communities. Understanding this community strategy to contain infections within the community has helped health care workers to apply effective measures to cope with other infectious diseases.

This study aimed to elucidate the implementation of boat quarantine in response to the SARS-CoV-2 pandemic within the fishing communities in Trat province, Thailand. The results showed that under certain circumstances, an outbreak can be successfully controlled through a model of community quarantine. The SARS-CoV-2 prevention and control measures that the communities implemented aimed to assess the risk among fishermen and reduce the spread of disease that put pressure on health services. In this paper, the term ‘boat quarantine’ refers to the use of a fishing boat as a ‘quarantine’ and ‘isolation’ place for those who are infected with SARS-CoV-2, in order to keep infected individuals away from each other in order to lower the chances of spreading the virus. The term ‘boat’ is used in this paper instead of ‘ship’ or ‘vessel.’

## 2. Methods and Data Collection

The SARS-CoV-2 prevention and control measures consisted of boat quarantine for members of the fishing communities in Trat province. In order to understand the implementation of boat quarantine for SARS-CoV-2, this study employed qualitative research methods, as these methods allow for participants to describe their experience of controlling the SARS-CoV-2 pandemic within their community. Qualitative research methods allow for researchers to understand the lives of the participants, as well as their experiences, as reported in their own words [18,19].

The researcher employed a purposive sampling technique to recruit the key participants, who are people involved in the control and prevention of SARS-CoV-2 among fishermen in the fishing communities in Trat province. The inclusion criteria comprised people who had worked to contain SARs-CoV-2 among fishermen for more than three months at the time of participation. The forty-five participants included ten fishermen, ten health care officers, eight health care volunteers, three port managers, three port-based officers, three maritime enforcement commanders, three border control officers, three local fishery network professionals, and two individuals working for an NGO (Table 1). This data collection focused on fishermen working in distant waters (not artisanal fisheries). The number of participants was determined using a theoretical sampling technique, which involves ceasing recruiting when little new data emerge, signifying data saturation [20,21]. The researcher conducted the initial data analysis concurrently with the data collection. This allowed for the researcher to be able to determine data saturation, as they could not identify any further codes and themes that could be developed. A boat in this paper refers to a motorized fishing boat that can contain approximately 10–20 crewmembers.

The data collection was conducted during the SARS-CoV-2 pandemic in 2021–2022 through in-depth interviews among the key participants, in response to SARS-CoV-2 prevention and control measures within the community, to elucidate their experiences with boat quarantine. Prior to the commencement of the study, ethical approval was obtained from the researchers’ institution. Prior to making an appointment for the interviews, the participants’ consent to participate in the study was sought. Following a full and detailed explanation of the study, the length of interviewing time, and the scope of questions, the participants were asked to sign a consent form, which was kept in a locked filing cabinet to protect the confidentiality of the participants. They were assured that all data would be kept confidential and pseudonyms would be used.

With permission from the participants, interviews lasting between 40 and 60 min were tape recorded. Some gifts were provided to the participants in thanks for taking part in this study. The interviews consisted of open-ended questions about SARS-CoV-2 control and prevention measures within the fishing communities. Interviews were conducted following an interview guide that included the following topics: SARS-CoV-2 prevention and control, the SARS-CoV-2 situation, and boat quarantine.

The in-depth interviews were analyzed to identify participants’ experiences with boat quarantine. A thematic analysis was used to identify key ideas and concepts from the interviews which aimed to identify, analyze, and report patterns or themes within the data (Figure 1). All transcripts were produced within 72 h of the interviews being completed. The tapes were then transcribed verbatim in Thai for data analysis. Initially, the researcher performed open coding once codes were first developed and named. Then, axial coding was applied to develop the final themes within the data. This was carried out by re-organizing the codes that the researcher had developed from the data during open coding, to identify new ways by making connections between categories and sub-categories. The researcher read the text to become familiar with the data and then organized the management process of prevention and control measures of SARS-CoV-2 among fishermen, which later became the themes of the study.

In terms of rigor, this study used triangulation methods to ensure that the data were sufficiently trustworthy. In-depth interviews and participant observations were used for data collection. Participant observations were used as a means of field observations. The researcher participated in SARS-CoV-2 testing among fishermen.

### Participant Observation

This boat (A) was used as a quarantine place for 14 crewmembers who remained onboard for 14 days. There were 11 Cambodian workers and 3 Thai workers on the boat: a boatswain, skipper, and captain. All fish could be thrown overboard on the dock, and the vessel was washed down every day while the boat lay in the quarantine port. Once all the fish were unloaded at the port, all of the crewmembers were sent to the medical tent set up nearby, where a public health care team and governmental officials provided them a SARS-CoV-2 test. While they were waiting for the results, they remained in a restricted area. The dock, boat, and adjacent buildings were considered as the restricted area (red zone). The access gateway was guarded by officers. Entry into the red zone required personal protective equipment (PPE). The results indicated that four crewmembers tested positive. These crewmembers were taken to the hospital immediately for further treatment. The rest were asked to quarantine onboard. The officers asked the captain to relocate the boat to the governmental port where the boat would remain for 14 days. During the quarantine period, the boatswain purchased food and supplies for the fishermen while they stayed on the boat for isolation. The officers visited the boat to re-test the crewmembers for SARS-CoV-2. Most of the crew showed no sign of sickness.

## 3. Findings

The findings in this study are intended to provide a basis for addressing the lesson learned by implementing boat quarantine for fishermen. The information obtained from these interviews offers insight into the relatively important process of controlling SARS-CoV-2 onshore and onboard.

### 3.1. Assessing the Risk of COVID-19 Infection

Boats are highly susceptible environments for the rapid spread of infectious diseases due to the living conditions on boats. Sick crewmembers are likely to have been in contact with almost all of the crew. A health care officer and port officer explained:

“Most fishes caught in the Gulf of Thailand are caught through the small-scale commercial fishing sector. We considered a boat as a community, so we applied the concept of community quarantine there following the COVID-19 guidelines.”

In this case, all of the crewmembers had to quarantine for 14 days due to close contact, according to local advice. Boat quarantine was followed by a 14-day government-regulated quarantine and further testing. The health care officers ensured continued health care support for the seafarers onboard. A health care officer and a pier officer spoke about the SARS-CoV-2 situation among fishermen:

“Our plans and procedures changed according to the number of new infections within the community. When there was a rapid spread of COVID-19 among fishermen, the health care officers set up a testing center near the port. This made it easy to test and treat the patients.”

“The testing can only provide a snapshot of the moment, but it cannot predict whether the virus will develop into COVID-19 in the coming days. Thus, quarantining was a must. We could not find a place to quarantine. Then, one of the boat owners suggested why not use a boat as a place to quarantine.”

The port manager said:

“Staying at the port for a period of time made it risky for workers to maintain contact with family and friends. During the outbreak, the fishermen with signs or symptoms, which are contagious, but who did not have the disease, quickly infected others.”

Two of the health care officers mentioned the reason for setting up boat quarantine as follows:

“In 2019 to early 2020, we could control the disease. It was controllable at that time until the middle of 2020, when there was an outbreak among fishermen and their family members. The fishermen had been infected with COVID-19, and the disease was widespread in the community. There was not enough space for quarantine. Then, one of port officers asked in a meeting, why not use boats as quarantine facilities? Then, we start the discussion on using boats.”

“During that time, there were many people who tested positive. One factory found many hundreds of patients. We could not find isolation and quarantine places for all of them. Moreover, there were many fishermen who tested positive. So, we decided to use the boats as a place for isolation and quarantine under the care of health officers.”

### 3.2. Shipboard Measures to Address Risks Associated with SARS-CoV-2

All fishermen tried to protect themselves from SARS-CoV-2. The crewmembers were subjected to the usual procedures, including temperature checks and testing. They all wore well-fitting masks during operations both at sea and at the pier. While they were on duty, the crewmembers had to self-monitor for symptoms and report anything suggestive of SARS-CoV-2 immediately to the crew leader. Daily temperature screenings were performed by the crew leader. They demonstrated no elevated temperature or symptoms of acute respiratory infection. The captain described the SARS-CoV-2 prevention measure when they were onboard:

“I have prepared all of the medical equipment. All of the crew is required to have daily temperature check-ups. Although wearing masks was necessary, it was impossible to do so at sea. All of the crewmembers shared a bedroom, which was small and tidy. If one of them was infected with COVID-19, it means all of them must also be infected.”

During the operation, the captain was responsible for reporting all suspect cases to the relevant health authorities at the port. They had to keep track of their symptoms and vital signs. At sea, if any of the crewmembers showed signs and symptoms suggestive of a possible SARS-CoV-2 infection, the captain was obligated to report it to the medical officer on land immediately. The early recognition and close monitoring of a case was the responsibility of the crew leaders. Assistance with making a diagnosis for fishermen was available, as the health care officers provided further advice for the management of the cases by telephone. If there were sick crewmembers, the work was called off and the boat had to return to the pier. A port authority mentioned the monitoring system at sea:

“I received a call from the captain once that one of the crewmembers showed signs of COVID-19. So I asked them to return to the port immediately. The crewmember who was sick had to be tested for COVID-19 and the result turned out positive. He was taken to the hospital. Then, all of crewmembers had to be quarantined on the boat. A few days later, when the health care professional retested, the results turned out positive.”

“On the ship, if one fish worker showed signs of COVID-19 symptoms, all crewmembers assumably came into close contact and should therefore quarantine for 14 days even though half of them tested negative. They received care from health care workers, just like those who quarantined within the community.”

### 3.3. Managing Suspected COVID-19 Cases and Their Contacts

Due to the pandemic, the fishermen were confined to COVID-19 restrictions while in port, or their access to the community was highly restricted. Although most fishermen were asymptomatic, all crewmembers had to take PCR tests. If the result turned out positive, they were referred to the hospitals or isolation within the community. They had to quarantine for 14 days if they were exposed to SARS-CoV-2. If one of the crewmembers was infected with the disease, the rest were obligated to isolate onboard. The fishing vessels went directly to the fishing port, which is under private ownership.

“Upon embarking, all crewmembers must be tested immediately. If one or more crewmembers test positive, they all have to quarantine on boats. During quarantine, the health care officers watched for symptoms for 14 days after they had come into close contact with the crewmember with COVID-19.”

They had to be tested upon arrival at the dock. Prior to the beginning of the quarantine, all crewmembers were asked to have SARS-CoV-2 tests; the positive disembarked and were transferred to the hospital. The rest of the crew had to stay onboard for quarantine. A port authority described the process:

“Prior to the boat’s arrival at the port, the captain would tell us the approximate date and time of arrival. I would then confirm with the health care officers. Then, when the boat arrived at the port, they were only allowed to unload the fish and stay in the restricted area. The community bus would take all of the crewmembers to the COVID-19 center for testing. Then, the crewmembers had to wait for the results on the boat. If the results turned out positive, they all had to be isolated on the boat.”

A worker with a positive test was transferred to the hospital immediately. The officers allowed the fishermen to continue fishing as they perceived those fishermen work and live onboard with limited access to social contact and activities. The maritime enforcement commander said:

“The fishermen usually stayed on the boat at least for 14 days so we allowed them to quarantine onboard and also fish at sea. They may have the disease but did not show symptoms, or they were exposed to the disease but did not know it. They could not just on stay on the boat for 14 days and do nothing. The boat owners had to pay their salary although they had not carried out any fishing.”

### 3.4. Shipboard Quarantine

In terms of SARS-CoV-2, the fishermen had to quarantine and stay away from others when they had been in close contact with someone who had SARS-CoV-2, while those people who tested positive for SARS-CoV-2 had to be isolated, even if they did not have symptoms. Their restrictions could be linked to community protocols. Importantly, the 14-day quarantine at the point of embarkation was the best way to ensure that the crewmembers did not transmit SARS-CoV-2 to the community. Crewmembers who had been at sea with no contact with the outside world for 14 days are assumed to be ”free of COVID“ the health care officer said,

“The good thing about being in quarantine on a boat was that they could access medical treatment ashore and be taken to shore in a timely manner for treatment in the event of serious illness.”

Importantly, quarantine is a strategy to prevent the transmission of SARS-CoV-2 by keeping people who have been in close contact with someone with SARS-CoV-2 apart from others, as the time from initial contact with the virus until symptoms start ranges from 1 to 14 days; thus, the incubation period is considered to be 14 days. Infection surveillance involved retesting for SARS-CoV-2 on the first day, seventh, and last days of quarantine. The advantage of quarantine relies on the early detection of cases by monitoring people who may have come in contact with the infection.

The fishermen were confined to the boat and were in strict isolation, and they could only stay on the boat. The boat was separated into areas that reflected the level of risk of contamination and infection. During quarantine, food and water resources were provided by the boat owner. The boat owners helped their workers by buying items that helped to make their lives more comfortable during the onboard quarantine. One of the boat owners said:

“Two of my Cambodian crewmembers were infected with COVID-19. They were transferred to the hospital, whereas the rest were quarantined onboard. I knew that staying onboard for 14 days would be miserable, so I brought them some food, water, and games. I also bought phone cards for them so that they could call their family and use the internet onboard.”

Quarantine was carefully monitored by the authorities, and penalties for breaking quarantine were stiff. The restrictions were imposed by port authorities who were concerned that their workers could be exposed to SARS-CoV-2 while onshore.

“One night, I was called to come to the port urgently as one of the crew had jumped off the boat. He swam from the sea to the port. They were isolated on the boat as one of his team had tested positive. He said that he missed his family who lived in the village, so he decided to swim to land.”

### 3.5. Implementing Infection Prevention and Control Measures

All participants agreed that the most effective way to prevent SARS-CoV-2 was vaccination. In early 2021, the availability of vaccination supplies was limited. The SARS-CoV-2 vaccine priority was the high-risk population, especially the elderly. The private sector used the Sinopharm vaccine when supply was limited:

“At first, the vaccine supply was limited, and it was distributed to the elderly. There were no COVID-19 vaccines that we could provide for seafarers. Hence, the fishery business sector paid for the vaccine for the seafarers. The business sectors have to go on and we could not wait anymore. Soon, the vaccine from the government will be available for migrant workers, then all seafarers will have been injected.”

The Thai government started vaccinating Thais and legal foreign workers on fishing vessels in 2021, citing the importance of the fishing industry to their economic well-being. The aim was to have as many seafarers as possible register, which in turn would allow them access to the vaccines. Health care officers allocated the supply of the vaccine to all crews, no matter if they were Thais or foreigners.

“We tried to organize mass vaccination among seafarers, no matter if they were Thai or Cambodian. At the beginning, they bought the Sinopharm vaccine. I believed that vaccination was an effective strategy for mitigation among seafarers.”

## 4. Discussion and Conclusions

The deadly infectious SARS-CoV-2 disease spread worldwide, so effective strategies have been required to lessen the risk. The public health care measures to protect and prevent exposure to contagious diseases involves ‘quarantine measures’, which are a way to separate and restrict the movement of people who are exposed to a contagious disease, and to determine whether they become sick or not. The SARS-CoV-2 pandemic had an impact on fishing management and fishermen. The model of boat quarantine has been used for many years [22]. Moreover, boat quarantine is an effective model in response to outbreaks and this model has the potential to solve the problem of staffing shortages and to maintain the supply chains of local businesses, as well as other systems that are essential for maintaining a well-functioning society and economy. This study showed that under certain circumstances, an outbreak can be successfully controlled by implementing boat quarantine.

There is considerable danger in contracting diseases in ports where trade (selling fish) by boat occurs in SARS-CoV-2 infected regions [23]. In these fishing communities, the sea is a natural barrier and is used to separate unhealthy people from the healthy. Fishing operations are inherently certain, so that the surveillance model needs to be flexible to ensure the quality of life of the fishermen while they stay on the boat. Boat quarantine is effective in terms of monitoring and controlling who is actually onboard a fishing boat. However, to implement community quarantine measures, there is a need to enhance public health risk communication activities both at the information and the communication levels [3].

During periods of critical staffing shortages due to self-isolation, boats could be considered as a quarantine facility. Largely due to economic and social concerns, the pandemic has also had an impact on the local fishing communities. The study emphasizes that the testing of arriving crews requires cooperation from many stakeholders. These restrictive measures are imposed by government and port authorities as they perceive that seafarers on fishing vessels could be a potential source of infection.

Notably, there are no guidelines for boat quarantine. There have been no efforts to minimize psychological stressors during quarantine [24]. Wong [25] noted that crew change crises, low vaccination rates, and the lack of key worker recognition can impact the mental health of seafarers with regards to SARS-CoV-2. Given these challenges, it is important for fishermen to be involved in designing an infection control system. Although other researchers found that working at sea made it difficult for seafarers to access health care, the seafarers in this study were able to reach a health care advisor through communication networks at any time. Drawing on these insights, the local public and private sectors provided more responsive and targeted interventions to address infections within fishing communities. The port authorities and boat owners became important allies with policy makers that helped draw attention to their situation.

The advantage of using boat quarantine is that it can be implemented in the event of a shortage of health care facilities. Moreover, a successful initiative on boat quarantine involves the close collaboration of public and private sectors. Thus, there is a need to strengthen community quarantine measures that involves cooperation among community members and stakeholders for surveillance, supporting outreach workers, knowledge sharing and management, and public health risk assessments and response coordination.

## 5. Limitation

The limitation of this study is that it focuses only on boat quarantine in Trat province. Boat quarantine in other provinces may vary due to the socio-political contexts of each setting. This study reflects the practices only of fishermen in Trat province and does not systematically distinguish the vessel size and specific gear technologies. The use of boats for quarantining took into consideration the context and characteristics of the boat.

## Figures and Tables

**Figure 1 ijerph-20-04816-f001:**
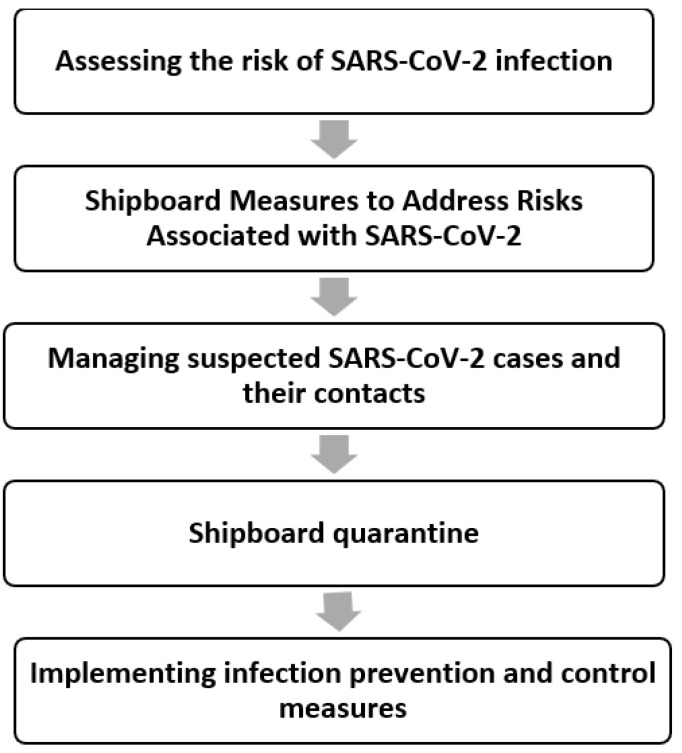
Guidance for implementing SARS-CoV-2 prevention and control strategies in the context of fishing communities.

**Table 1 ijerph-20-04816-t001:** Socio-demographics of the participants (n = 45).

Characteristics		No.
Age	30–39	11
	40–49	27
	50–59	7
Sex	Male	38
	Female	7
Marital status	Single	5
	Married	31
	Divorced	9
Employment	Government official	9
	Health care officers	10
	Health care volunteers	8
	Fishery Businessmen	6
	Fishermen	10
	NGOs	2
Education	Junior high school	6
	High school	5
	Vocational School	5
	Undergraduate	29

## Data Availability

Data available on request due to ethical restrictions.

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
