# Peer review of "‘Boat Quarantine’: Lessons Learned from SARS-CoV-2 Prevention and Control Measures in Fishing Communities in Thailand"

_ijerph, 2023, doi:10.3390/ijerph20064816_

Round 1

Reviewer 1 Report

In this manuscript “Boat Quarantine’: Lesson learned from the COVID-19 prevention and control in the fishing communities, Thailand”  Niphattra Haritavorn lays out how quarantining in boats can be a tool against SARS-CoV-2 and possibly other pandemic causing infections. The study was confine to a single region in Thailand with a limited number of subjects.   In the manuscript the author describes:  Assessing the risk of COVID-19 infection, Shipboard Measures to Address Risks Associated with COVID-19, Managing suspected COVID-19 cases and their contacts, Shipboard quarantine Implementing infection prevention and control measures.

Overall, the manuscript must be rewritten to enhance readability.  The manuscript was written with flowery language.  The manuscript needs to be written in a scientific language.  Throughout the manuscript the author uses COVID-19.  I think the author means SARS-CoV-2. This must be changed throughout the manuscript.   

The use of English language must be enhanced.  Throughout the manuscript the authors do not use correct English.  For example, line 336 the authors state “There is considerable danger of contacting the disease in ports where trade (selling fish) by boat occurs with COVID-19 infected regions [21]”   I think the authors mean contracting and NOT contacting.  The reviewer is not exactly sure what this sentence means. Also, this sentence and many other sentences like this one throughout the manuscript must be revised.

Throughout the manuscript the author put sentences together which there is no logical theme or these sentences do not make sense. For example, lines 345-350 the authors mention “While the boat is a closed environment, the sea is open space. During periods of critical staffing and isolation shortages, boats could be considered as a quarantine facility. Largely due to economic and social concerns, the pandemic has also had an impact on the local fishery community. The study emphasizes that testing of arriving crews required cooperation from many stakeholders. These restrictive measures are imposed by government and port authorities as they perceived the seafarers on fishing vessels could be a potential source of infection.”  Here, the authors mention “While the boat is a closed environment, the sea is open space.”  This is an obvious fact and there is no reason to mention it but has nothing to do with the next sentence or the rest of the paragraph.

Author Response

Comment 1: Throughout the manuscript the author uses COVID-19.  I think the author means This must be changed throughout the manuscript.   

Response1: Throughout the paper, the author has changed the term ‘COVID-19’ to ‘SARS-CoV-2,’ except the quoting that the participants used the term ‘COVID-19’.

Commment2: The use of English language must be enhanced.  Throughout the manuscript the authors do not use correct English.  For example, line 336 the authors state “There is considerable danger of contacting the disease in ports where trade (selling fish) by boat occurs with COVID-19 infected regions [21]”   I think the authors mean contracting and NOT contacting.  The reviewer is not exactly sure what this sentence means. Also, this sentence and many other sentences like this one throughout the manuscript must be revised.

Response2:The author revised the paper and submitted the paper to MDPI English editing service.

Comment3:

Throughout the manuscript the author put sentences together which there is no logical theme or these sentences do not make sense. For example, lines 345-350 the authors mention “While the boat is a closed environment, the sea is open space. During periods of critical staffing and isolation shortages, boats could be considered as a quarantine facility. Largely due to economic and social concerns, the pandemic has also had an impact on the local fishery community. The study emphasizes that testing of arriving crews required cooperation from many stakeholders. These restrictive measures are imposed by government and port authorities as they perceived the seafarers on fishing vessels could be a potential source of infection.”  Here, the authors mention “While the boat is a closed environment, the sea is open space.”  This is an obvious fact and there is no reason to mention it but has nothing to do with the next sentence or the rest of the paragraph.

Response3:The author deleted the irrelevant sentences and submitted the paper to MDPI English editing service.

Reviewer 2 Report

Quarantine is one of the most important and widely used non-pharmaceutical interventions during COVID-19 pandemic. It’s interesting to learn lessons in special environment like fishing communities described in the article.

1. Did authors follow qualitative research standards like COREQ?

2. The results mostly focused on the reasons and methods used for the “boat quarantine”. However, readers may also want to know the advantages and challenges using these offshore facilities.

3. A public health intervention could be quickly changed according to rapidly changing situation. Was there any change for the details during implementation? For example, is the quarantine period always 14 days? Or getting shorter when entering omicron era? Moreover, I think nowadays this kind of quarantine might already be canceled. If yes, how and when it finished mission?

Author Response

Comment 1: Did authors follow qualitative research standards like COREQ?

Response 1: No, this research followed the SRQR and the qualitative guideline from the following book. Liamputtong, P., Qualitative research methods. 2013, Melbourne: Oxford University Press.

Comment 2: The results mostly focused on the reasons and methods used for the “boat quarantine”. However, readers may also want to know the advantages and challenges using these offshore facilities.

Response 2:  The advantage of using boat quarantine is that it can be implemented in the event of a shortage of health care facilities. Moreover, a successful initiative on boat quarantine involves the close collaboration of public and private sectors. Thus, there is a need to strengthen community quarantine measures that involves cooperation among community members and stakeholders for surveillance, supporting outreach workers, knowledge sharing and management, and public health risk assessments and response coordination.

Comment 3: A public health intervention could be quickly changed according to rapidly changing situation. Was there any change for the details during implementation? For example, is the quarantine period always 14 days? Or getting shorter when entering omicron era? Moreover, I think nowadays this kind of quarantine might already be canceled. If yes, how and when it finished mission?

Response 3:  Due to the shortage of health facilities between September and December 2021, the peak time of the pandemic, boat quarantine was implemented. The quarantine followed the Ministry of Public Health’s SARS-CoV-2 (COVID-19) guidelines that stated that the official quarantine period was to last 14 days. 

Reviewer 3 Report

1. The objective(s) is/are not clearly specified

2. The author needs to explain what is the need for this study

3. Sample size is too small

4. Descriptive manuscript

Author Response

Comment1: The objective(s) is/are not clearly specified

Response1: This study aimed to elucidate the implementation of boat quarantine in response to the SARS-CoV-2 pandemic within the fishing communities in Trat province, Thailand.

Comment2: The author needs to explain what is the need for this study

Response2:  The following sentences have been added, "boat quarantine serves as a good model to study the potential for SARS-CoV-2 to spread within a population in the fishing communities. Understanding this community strategy to contain infections within the community has helped health care workers to apply effective measures to cope with other infectious diseases."

Comment3: Sample size is too small

Response3: The number of participants was determined using a theoretical sampling technique, which involves ceasing recruiting when little new data emerge, signifying data saturation (Liamputtong, 2013; Patton, 2015). The researcher conducted the initial data analysis concurrently with the data collection. This allowed for the researchers to be able to determine data saturation, as they could not identify any further codes and themes that could be developed.

Comment 4: Descriptive manuscript

Response 4: Because this study aimed to explore the experience of controlling the SARS-CoV-2 pandemic within their community,  descriptive phenomenology was adopted as methodological framework. Descriptive phenomenology allows researcher to understand the issues under study from the experiences of those who have lived through them (Carpenter, 2013)

Round 2

Reviewer 1 Report

The authors responded correctly to the reviewer's comments.